# The importance of multiple scatterings
# in medium-induced gluon radiation

**Carlota Andres[1], Fabio Dominguez[2] and Marcos Gonzalez Martinez[2]⋆**

**1** CPHT, CNRS, Ecole Polytechnique, IP Paris, F-91128 Palaiseau, France
**2** Instituto Galego de Física de Altas Enerxías IGFAE, Universidade de Santiago de Compostela, E-15782 Santiago de Compostela (Galicia), Spain

⋆ marcosg.martinez@usc.es

## Abstract

In this work we disentangle the underlying physical picture of the in-medium gluon radiation process across its different energy regimes by comparing the recently obtained fully-resummed – without any further approximations – BDMPS-Z in-medium emission spectrum with the extensively used analytical approaches. We observe that in the high-energy regime the radiation process is dominated by a single hard scattering, while in the intermediate-energy region coherence effects among multiple scatterings are crucial. Finally, we prove that in the low-energy regime the dynamics is again controlled by a single scattering but where one must include a suppression factor accounting for the probability of not having any further scatterings.

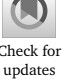
## 1 Introduction

The single-inclusive gluon emission spectrum is the building block of medium-induced radiation. Usually, this spectrum is computed within analytic approximations such as the Harmonic Oscillator (HO), where multiple in-medium scatterings are resummed within a Gaussian approximation, or the opacity expansion, which performs a series expansion on the number of scatterings. In this talk, based on [1], we make use of a recent approach which allows us to perform the numerical evaluation of the BDMPS-Z spectrum with full resummation of multiple scatterings without any further approximations [2] to determine the range of validity of the usually employed first opacity and HO results. Furthermore, we obtain the low-energy asymptotic limit of the fully resummed result, showing that in this regime the spectrum shall be interpreted as a single in-medium scattering times the probability of not having any further scatterings.

## 2 Medium-induced energy distribution

As we showed in [1], the low-energy asymptotic limit of the all-order medium-induced energy distribution off a hard parton traversing a brick of density $n_0$ and length $L$ is given by

$$\omega \frac{dI^{\text{med}}}{d\omega}\bigg|_{\omega \to 0} = \frac{2\alpha_s C_R}{\omega} \Re \int_0^L ds\, n_0 \int_0^s dt \int_{\vec{p}\vec{q}} i\, \frac{\vec{p}\cdot\vec{q}}{\vec{q}^2} \sigma(\vec{q}-\vec{p})\, e^{-\left(i\frac{p^2}{2\omega}+\frac{1}{2}n_0\Sigma(p^2)\right)(s-t)}, \quad (1)$$

where $\omega$ is the energy of the emitted gluon, $\vec{p}$ and $\vec{q}$ denote transverse momenta, $\sigma$ is the dipole cross section which can be written in terms of the collision rate $V$ as

$$\sigma(\vec{q}) = -V(\vec{q}) + (2\pi)^2 \delta^{(2)}(\vec{q}) \int_{\vec{l}} V(\vec{l}), \quad (2)$$

and $\Sigma$ is given by[1]

$$\Sigma(p^2) \equiv \int_{q^2>p^2} V(\vec{q}). \quad (3)$$

Clearly this expression has the form of the first opacity result ($N = 1$ GLV) times a no-scattering probability factor. This suppression factor comes from the resummation of the virtual contributions, and thus highlights the importance of accounting for multiple scatterings in order to properly describe the fully resummed result for low gluon energies. We illustrate this in Figure 1, where it can be seen that the all-order evaluation (solid lines) is well described by the asymptotic result given by eq. (1) (dotted lines) for different medium densities, while the first opacity approximation (dash-dotted) overpredicts the energy spectrum.

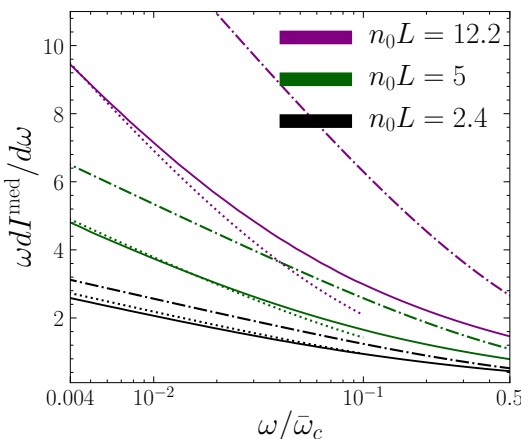

Figure 1: Fully resummed (solid lines), low energy limit given by eq. (1) (dotted), and first opacity (dash-dotted) in-medium energy spectra for a Yukawa parton-medium interaction as a function of $\omega/\bar{\omega}_c = 2\omega/\mu^2 L$ ($\mu$ being the screening mass of the Yukawa potential) for several values of $n_0 L$. Figure extracted from [1] under Creative Commons Attribution License (CC BY 4.0).

Moving to higher energies, multiple soft scatterings are expected to play an essential role in the intermediate-energy regime. We thus perform in this kinematic region a comparison between our all-order result and the so-called Improved Opacity Expansion (IOE) [3], an analytic approximation including multiple scatterings. This comparison is shown in Figure 2,

---

[1]See ref. [1] for further details.

where we plot the all-order energy distribution together with the IOE (HO+NLO) and single hard scattering (GLV $N = 1$) spectra for different medium densities. It can be clearly seen in this figure that the HO+NLO result agrees the full evaluation, while the single hard scattering approximation does not, thus proving that the inclusion of coherence effects among multiple scatterings is fundamental to properly describe the in-medium radiation process in this regime.

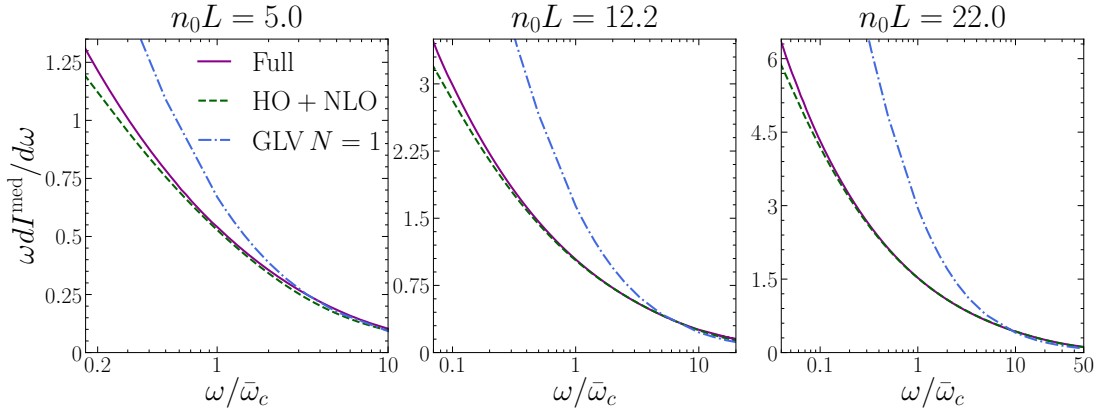

Figure 2: All-order (solid lines), IOE HO+NLO (dashed), and GLV $N = 1$ (dash-dotted) medium-induced gluon energy spectra for a Yukawa parton-medium inter-action as a function of $\omega/\bar{\omega}_c = 2\omega/\mu^2 L$ for different values of $n_0 L$. Figure extracted from [1] under Creative Commons Attribution License (CC BY 4.0).

Finally, we can see also in Figure 2 that for high gluon energies both the full and IOE results agree with the $N = 1$ GLV approximation, showing that, as expected, in this regime the spectrum is dominated by just one single hard scattering.

## 3 Conclusions

In this talk we present a comparison between the fully resummed medium-induced gluon radiation spectrum and the widely employed analytical approximations in order to discern the dominant dynamics across the different energy regimes. We find that in the high-energy regime the radiation process is dominated by a single hard scattering, while in the intermediate-energy region coherence effects among multiple scatterings become essential. Furthermore, we show for the first time that in the low-energy regime (also known as Bethe-Heitler regime) the energy distribution can be interpreted as the single scattering result times a probability of not having any further scatterings, thus, proving that multiple scattering effects are crucial to correctly describe the emission process in this region.

## Acknowledgements

**Funding information** This work was supported by Ministerio de Ciencia e Innovación of Spain under project FPA2017-83814-P; Unidad de Excelencia María de Maetzu under project MDM-2016-0692; Xunta de Galicia under project ED431C 2017/07; Consellería de Educación, Universidade e Formación Profesional as Centro de Investigación do Sistema universitario de Galicia (ED431G 2019/05); European Research Council under project ERC-2018-ADG-835105 YoctoLHC; and FEDER. C.A. was supported through H2020-MSCA-IF-2019 893021 JQ4LHC. M.G.M. was supported by Ministerio de Universidades of Spain through the National Program

FPU (grant number FPU18/01966).

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
