# Peer review of "The importance of multiple scatterings in medium-induced gluon radiation"

_SciPost Physics Proceedings, doi:SciPost Phys. Proc. 10, 016 (2022)_

## Round 2 · Referee Report · Anonymous (Referee 1) · 2022-1-28

Report

In this contribution to the ISMD2021 proceedings the authors compare their recent resummed result for the medium induced gluon bremsstrahlung spectrum to widely used analytical approximations. The manuscript is clearly written and meets the requirements. I only have two small suggestions for the benefit of non-expert readers that the authors may want to consider provided it is possible within the page limit. 1) explain in a few words what is meant by opacity expansion 2) after eq. (1) briefly say what p and q are and include the definition of Sigma

---

## Round 3 · Referee Report · Anonymous · 2022-2-16

Strengths

The presentation is clear and well-written, and comprehensible to a non-expert. Comparisons and highlighting of agreement and differences with respect to pre-existing approximations are very clear.

Weaknesses

The general context of the work and the inputs to the calculations are not clear to a non-expert, but this is not a major problem for the intended purpose, and the relevant introductory material is cited.

Report

A well-presented and clear report on an interesting development, with good comparison to previous methods.

---

## Editorial Decision

published